# Ultra Sensitivity Silicon-Based Photonic Crystal Microcavity Biosensors for Plasma Protein Detection in Patients with Pancreatic Cancer

**DOI:** 10.3390/mi11030282

**Published:** 2020-03-09

**Authors:** Chun-Ju Yang, Hai Yan, Naimei Tang, Yi Zou, Yas Al-Hadeethi, Xiaochuan Xu, Hamed Dalir, Ray T. Chen

**Affiliations:** 1Microelectronics Research Center, Electrical and Computer Engineering Department, University of Texas at Austin, Austin, TX 78758, USA; yanhai12345@gmail.com (H.Y.); firstround99@gmail.com (Y.Z.); coulee.xu@gmail.com (X.X.); 2Biosensor Division, Omega Optics Inc., Austin, TX 78757, USA; naimeit@gmail.com (N.T.); hamed.dalir@omegaoptics.com (H.D.); 3School of Information Science and Technology, ShanghaiTech University, Shanghai 201210, China; 4Physics Department, Faculty of Science, King Abdulaziz University, Jeddah 21589, Saudi Arabia; yalhadeethi@kau.edu.sa; 5State Key Laboratory on Tunable Laser Technology, Harbin Institute of Technology, Shenzhen 518055, China

**Keywords:** biosensor, photonic crystal, nanophotonics, cancer, silicon photonics

## Abstract

Defect-engineered photonic crystal (PC) microcavities were fabricated by UV photolithography and their corresponding sensitivities to biomarkers in patient plasma samples were compared for different resonant microcavity characteristics of quality factor Q and biomarker fill fraction. Three different biomarkers in plasma from pancreatic cancer patients were experimentally detected by conventional L13 defect-engineered microcavities without nanoholes and higher sensitivity L13 PC microcavities with nanoholes. 8.8 femto-molar (0.334 pg/mL) concentration of pancreatic cancer biomarker in patient plasma samples was experimentally detected which are 50 times dilution than ELISA in a PC microcavity with high quality factor and high analyte fill fraction.

## 1. Introduction

For early high-value bio-marker detection, a sensor with the highest sensitivity is desired. A sensor with a high sensitivity together with lab-on-chip miniaturization capability enables the early detection of multiple biomarkers with a minimum sample volume. Among all detection techniques, label-free assays are particularly promising because they not only avoid the complex chemistries caused by steric hindrance of labels but also reduce the cost associated with labeling. A label-free assay detects the specific binding between the target receptor of interest and a specific probe biomolecule via optical, mechanical or electrical transduction mechanism. In addition to the freedom from electromagnetic interference, optical detection techniques are better than other transduction methods due to their high sensitivity and provide better multiplexing capability. In recent years, various integrated optical devices have been developed for label-free bio-sensing [1,2,3,4], such as surface plasmon devices [5,6,7], microring resonators [8,9,10], silicon nanowires [11], nanoporous silicon waveguides [12], Mach-Zehnder interferometer (MZI) [13,14] and photonic crystal (PC)microcavities [15,16,17,18,19]. The PC devices demonstrated the highest sensitivities among existing chip-integrated nanophotonic sensing technologies [20]. The two-dimensional PC microcavity, because of its compact size (of a few square microns in surface area), high throughput [21,22,23,24] and the highest sensitivity [25,26], have attracted significant interest in biosensing.

Pancreatic cancer is the fourth leading root of cancer-related death in the USA. There has been minimal progress with regard to cancer-specific outcomes in recent decades. Although effective therapies will undoubtedly change the natural history of the disease, effective biomarkers are a promising tool that will likely have a positive impact and will undoubtedly have an important role in the management of patients with pancreatic ductal adenocarcinoma (PDA) in the future [27]. There has been a recent explosion in the pancreatic cancer detection field with more than two thousand biomarker studies covering thousands of informative genes as candidate biomarkers. Unfortunately, current biomarkers and the affiliated early cancer detection methods suffer from low diagnostic sensitivity and specificity with poor multiplexibility for swift diagnosis. At the current stage, the early detection for potential pancreatic cancer markers is insufficient to provide accurate and timely diagnosis. In this article, we have developed a new silicon CMOS (complementary metal oxide silicon) compatible highly multiplexed biosensing platform using defect-engineered photonic crystal waveguide (PCW) based slow light effect with proven results on single lung cancer biomarker detection, and drug screening with highest sensitivity while keeping the required specificity. The devices are fabricated using silicon IC foundry (low-cost and mass producibility assured) and can detect up to sixty-four different biomarkers without labeling in the same chip simultaneously (1 mm × 10 mm). The innovations are high sensitivity coupled together with a myriad of high value pancreatic cancer biomarkers. We are able to contribute to highly sensitive detection of very low concentration of the biomarkers representing the earliest stage pancreatic cancer. In PCW resonator based sensors, our theoretical studies showed that analyte absorbance, equipment limited spectral noise and temperature noise, significantly contribute to the final achievable detection limit for all platforms [28]. From the device perspective, two parameters which significantly perturb the detection limits are resonance quality factor (*Q*-factor) and analyte fill fraction($f_{B}$). The high *Q* promotes the interaction time between the optical mode and the analyte while the larger guided mode volume results in larger fill fraction which is important for sensitivity enhancement. And both factors are created to generate higher sensitivity [25,28]. In addition, PCWs offer the exceptional characteristic of high group index and therefore slow photon propagation speed that significantly augments light-matter contact resulting in lowering the detection limit in coupled waveguide microcavity structures. We demonstrated experimentally in our side-coupled two-dimensional (2D) PC cavity-waveguide architecture, that the degree of the slow-down feature in the coupling waveguide results in enhanced light-matter interaction [29]. Over following experiments, we demonstrated experimentally 50 femto-molar (3.35 pg/mL) sensitivity to the detection of the specific binding of avidin to biotin with a L55 type PC microcavity (55 missing holes) [25]. The role of biomarker fill fractions was also explored simultaneously and an intricate design incorporating all fill fraction, factors of *Q* and slow light led to spotted concentrations of 1 femto-molar (67 fg/mL) for the specific uncovering of avidin binding to biotin [28]. Still, concerns have existed about the potential for high volume manufacturing of photonic crystal devices that have factually relied on electron-beam lithography to express the nanostructures. Furthermore, concerns also occur about the potential to sense the biomarkers in phosphate buffered saline (PBS) versus real patient samples. [28] lay down the basis of this type of PC cavity devices systematically with demonstration on standard biotin/avidin system, while our work focused on the demonstration of the high-sensitivity biosensing of real biomarkers in patient plasma samples. The binding dynamics vary with the conjugate pairs used in experiments and affect the detection limit.

In this paper, we experimentally demonstrate the detection of three plasma proteins in plasma samples from patients with pancreatic cancer and compare the detection sensitivity versus conventional enzyme-linked immunosorbent assay (ELISA). Two types of PC microcavities, with similar slow light effect and high quality factor resonances but different analyte fill fractions, were investigated. Devices were fabricated by using 193 nm UV photo-lithography, and the biomarkers were detected in patient plasma samples with varying dilutions down to 50 times from the original concentration.

## 2. Materials and Methods

### 2.1. Device Principles

The working principle of resonator based sensors relies on detecting the resonance wavelength or frequency shift in response to the refractive index changes caused by binding between probe receptor and target biomolecules. The response can be described by first order perturbation theory [30], in which the change in eigen frequency $\Delta \omega_{m}$ of the mth cavity mode can be expressed as:(1)$$\Delta \omega_{m}=\frac{\omega_{m}}{2}\frac{{{\langle E_{m}|\varepsilon_{l}|E_{m}\rangle }_{V}}_{liquid}}{{{\langle E_{m}|\varepsilon_{l}|E_{m}\rangle }_{V}}_{liquid+dielectric}}\frac{\Delta \varepsilon_{l}}{\varepsilon_{l}}\frac{1}{\upsilon_{g,m}}$$
(2)$$f_{B}=\frac{\omega_{m}}{2}\frac{{{\langle E_{m}|\varepsilon_{l}|E_{m}\rangle }_{V}}_{liquid}}{{{\langle E_{m}|\varepsilon_{l}|E_{m}\rangle }_{V}}_{liquid+dielectric}}$$
where $\upsilon_{g,m}$ is the group velocity of the mth mode at the frequency $\omega_{m}$, and $\Delta \varepsilon_{l}$ is the change in dielectric constant of the analyte from $\varepsilon_{l}$ upon perturbation. Equation (Equation 1) indicates that the resonance wavelength shift is directly proportional to the fill fraction $f_{B}$, defined as the ratio of electric field energy existing outside of a dielectric structure to the total, and it is inversely proportional to the group velocity.

We have previously demonstrated that resonances in L13 PC microcavities have high Q≈26,760 in silicon-on-insulator (SOI) structure as well as high biosensing sensitivity [19]. The higher Q in L13 PCMs is due to the combined effects of lower radiation loss as the resonance moves deeper into the photonic band gap compared to L3 PCMs that are studied conventionally, and the larger mode volume of L13 PCMs compared to L3 PCMs. The increased length enables larger overlap of the optical mode with the analyte leading to higher sensitivity [20]. Figure 1a shows the electric field profile, for a L13-type PC microcavities (13 missing air holes in the PC lattice). In silicon 2D PCs, light is coupled from a PC waveguide (PCW) into a PC microcavity. Isolated PC microcavities without PCW, in water ambient, are simulated to eliminate the coupling effect between PC microcavity and PCW, and focus on performance of the PC microcavity only. $f_{B}$ can be dramatically increased by introducing nanoholes into the PC microcavities [28] as shown in Figure 1b. The nanoholes are defined as air holes with radius $R_{D}$ smaller than the radius *R* of air holes in the bulk lattice. Nanoholes are introduced at the antinodes of the resonance mode of the L13 PC microcavity in Figure 1a. Previously, we showed that in a multi-resonance PC microcavity, such as the L21 and L55 [25] and the L13 PC microcavity studied here, the slow light effect in the coupling PCW contributes to the PC microcavity resonance sensitivity [29]. In multi-resonance PC microcavities, resonances closer to the transmission band edge, where the slow light effect is higher, have larger shift in the resonance wavelength for the same binding event, thus the same change in refractive index, and hence have larger sensitivity. We therefore consider the resonance mode that is closest to the PCW transmission band edge in our detection. As shown previously [28], after introducing PC microcavity with nanoholes, the analyte fill fraction $f_{B}$ can be significantly increased to 18% by suitable design with nanoholes versus approximately only 10% without nanoholes. It is however necessary to ensure that the enhanced analyte overlap does not compromise the resonance quality factor Q, which is another critical factor influencing PC microcavity resonance sensitivity [28].

L13 PC microcavities with nanoholes were fabricated with different nanohole radius $R_{D}$ = 0.4, 0.5 and 0.6 times the surrounding PCW hole radius *R*. Figure 2 shows the dispersion diagram of the guided mode in a PCW (indicated by a solid line) and the resonance frequency of the corresponding PC microcavity coupled to the PCW (indicated by a dashed line), obtained by three-dimensional (3D) plane wave expansion (PWE) simulations. The solid lines (blue, green, red) represent the guided mode in different waveguide width (W), while the dashed lines (blue, green, red) represent the resonance mode of the PC microcavities with different nanohole sizes. The cross points between the solid and dashed lines indicate the coupling between the waveguide and the microcavity, which corresponds to the resonances observed in the transmission spectrum (marked by arrows) in Figure 3. Simulations are done for both the L13 PC microcavity without nanoholes and L13 PC microcavities with different sized nanoholes. Light is confined in-plane by the periodic PC lattice and confined out-of-plane by total internal reflection at the core-cladding interfaces. The bottom cladding for vertical out-of-plane confinement is provided by the bottom silicon dioxide in SOI, while the ambient medium (water or phosphate buffered saline) serves as the top cladding. The ambient medium also fills the etched air holes in the triangular lattice in Figure 1a,b. Introduction of nanoholes in the PC microcavity raises the resonance mode frequency in the photonic band gap (PBG) as seen from Figure 2. It is therefore necessary to reduce the PCW width to ensure that the PCW guided mode is also raised in frequency in the PBG, so that the resonance mode of the nanohole cavity couples to the PCW guided mode in a wavelength range with high group index. Thus, when $R_{D}=0.4R$, the resonance mode (indicated by dashed blue line) couples under the light line to the W0.935 PCW (indicated by the solid blue line). The width of a standard W1 PCW is $\sqrt[{}]{3}a$ where a is the lattice constant. (W0.935 means the width of the PC waveguide is $0.935\sqrt[{}]{3}a$ where a is the lattice constant of the PC lattice). Furthermore, since the resonance mode and the waveguided mode are both raised in frequency in the PBG with increasing $R_{D}$, hence the PCW guided mode bandwidth progressively reduces from $R_{D}=0.4R$ to $R_{D}=0.6R$.

Figure 3a–c shows the experimentally obtained transmission spectra in L13 PC microcavity coupled waveguide devices with $R_{D}=0.4R$, 0.5*R* and 0.6R respectively. As observed from Figure 3, the respective PCW widths are denoted by W0.935, W0.895 and W0.865 respectively. The corresponding dispersion diagrams were shown in Figure 2. Multiple resonances are observed, however only the resonances indicated by black arrows closest to the PCW transmission band edge are of interest.

The bulk sensitivity increased by 66% from ≈68 nm/RIU for L13 [25] to ≈112 nm/RIU for L13 with nanoholes [28]. It shows that the wavelength shift sensitivity can be enhanced with increased nanohole size. As expected, since the resonance modes increase in frequency in the PBG with increasing $R_{D}$, the experimentally observed *Q* for the resonance closest to the transmission band edge, decreased with increased $R_{D}$ due to reduced fraction of the resonance mode energy below the light line for $R_{D}=0.6R$ [28]. In order to achieve high $f_{B}$ and highest *Q*, we selected L13 nanohole devices with $R_{D}=0.4R$ for further characterization and sensing.

### 2.2. Device Characterization

Subsequent to preliminary device characterization, all devices used in this research were fabricated from a commercial foundry using 193 nm UV photolithography. Figure 4a,b shows scanning electron micrograph (SEM) images of L13 PC microcavity devices, with and without nanoholes, fabricated by photolithography followed by reactive ion etching in Epixfab, Ghent, Belgium. The microcavities are fabricated in 220 nm silicon in a SOI wafer within a photonic crystal defined by a triangular lattice of air holes with hole radius R=112.5 nm. The nanoholes have radius $R_{D}=0.4R$ and are located in the lattice positions to coincide with the antinodes of the mode in the center of the L13 PC microcavity, as shown in Figure 1b. Although current Epixfab guidelines designate a minimum feature size of 150 nm, as observed from Figure 4b, we were able to achieve feature size of 90 nm in foundry fabricated devices, by judicious choice of CAD parameters, taking into account lithography dose-matrix effects. The corresponding transmission spectra are shown in Figure 5. Sharp resonances with *Q* = 22,000 and *Q* = 12,000 are achieved for the L13 PC microcavities, with and without nanoholes, as observed in Figure 5a,b, respectively in water ambient. Better quality factors in foundry fabricated samples are essentially indicative of better control of silicon etch conditions in a CMOS foundry than in a university cleanroom. Group index engineering, similar to our previous research was incorporated in all devices to enable high coupling efficiencies between the input and output strip waveguides and photonic crystal waveguides [16]. Sharp transmission band edges, characteristic of completely etched silicon air holes with vertical sidewalls, are also observed.

### 2.3. Functionalization

The sensor surfaces were cleaned in piranha solution $\left(H_{2}O_{2}:H_{2}SO_{4}=1:2\right)$ for 10 min to remove organic residual. The chips were next silanized with 2% by volume of 3-aminopropyl-triethoxy-silane (3-APTES) in toluene for 30 min followed by rinsing with toluene for 3 times. The washing steps are designed to remove the unbound molecules from the chip surface. Subsequently, the chips were cured in $110{\;}^{\circ }C$ for 30 min to mechanically stabilize the formation of the APTES films [31]. Then, the chip was incubated in 1.25% glutaraldehyde in phosphate buffered saline (PBS) for 1 hour and washed 3 times in PBS. The chips were then incubated with probe antibody in a humidity chamber and stored at $4{\;}^{\circ }C$ overnight. Following antibody immobilization, the chips were blocked with 1% bovine serum albumin (BSA) solution in PBS buffer (pH 7.3) in room temperature for one hour to block any binding sites that have not been covered by probe proteins. After a thorough triple wash in PBS, the devices were ready for test. The binding chemical structure in this process is shown in Figure 6.

### 2.4. Antibodies, Coupling Reagents and Derivatization

The antibodies or chemical sample we couple to the PC resonance cavities are shown in Table 1. Fasligand, chemokine ligand 4 (MIP1) and hepatic growth factor (HGF) are the antibodies used to detect corresponding antigens in the patient plasma sample. Human IgG antibody is employed for the negative control test to demonstrate that there is no reaction on our sensor for non-conjugate pairs.

### 2.5. Pancreatic Cancer Plasma Samples

Plasma samples from patients with pancreatic cancer were obtained from a case-control study of pancreatic cancer conducted at The University of Texas MD Anderson Cancer Center. Three protein markers, i.e. HGF, MIP1 and Fasligand were previously tested using ELISA. ELISA test result is shown in Table 3; the amount of antigen detected is presented in pico-gram and pico-mole. The molecular weights of antigen and antibody are also included for reference in Table 2.

## 3. Results and Discussion

The devices were first enclosed by a microfluidic well to control liquid volume and slow down evaporation. The dimensions of the microfluidic well, along the waveguide length, were chosen to ensure that the sub-wavelength grating couplers used to couple light from external optical fibers into the silicon photonic crystal waveguides are outside the well [32]. Before applying any target solution, the baseline resonance spectrum was recorded. The chip is then incubated in target solution for 40 min. Several concentrations of the target were measured on the same device. After each incubation, the chip was washed with PBS for three times and new spectra were tested and resonance positions were recorded. Experiments were first performed to detect the Fasligand biomarker in the patient plasma samples because of the relatively lower concentration compared with the other two proteins. Figure 7a shows the real-time data collected through the successive target addition and washing steps. Figure 7b shows the resonance in the transmission spectrum of the L13 PC sensor without nanoholes, for different dilutions of the patient plasma, by 50 times, 10 times, 5 times and 1 time (undiluted) respectively in PBS. A measurable red-shift of the resonance for increasing concentrations of target can be observed for the undiluted sample only. The resonance wavelength shifts, from baseline, were determined experimentally after each combination of target addition and washing for each dilution of the patient plasma samples.

The resonance wavelength change for each specific dilution is plotted in Figure 7c. The dashed curve in Figure 7c is at 0.04 nm representing the noise floor of our bio detection system, mainly from the +/− 0.02 nm wavelength imprecision of our optical spectrum analyzer (OSA). In our L13 photonic crystal microcavity devices, the temperature dependence of wavelength shift is measured to be 0.08 nm/${\;}^{\circ }C$. During measurements, it was confirmed that the chip temperature is stabilized to within +/−$0.1{\;}^{\circ }C$, the stage is thermally controlled with Newport 3040 Temperature Controller to avoid temper-ature induced resonance shift in the biosensors, thus the error from temperature is much lesser than our OSA detection error. Hence, only resonance wavelength shifts larger than 0.04 nm were treated as actual shifts caused by the specific binding between the probe and target. The L13 PC microcavity is employed to detect the Fasligand pancreatic cancer biomarker with the results illustrated in Figure 7c which is with 1 time dilution in the chosen patient sample with 16.73 pg/mL (0.44pM) concentration.

To verify the specificity of our test, Human IgG was introduced as a non-specific binding probe. The specificity was tested with undiluted pancreatic patient plasma sample. The resonance shift versus time was measured and plotted as shown in Figure 7d. It shows red shift in the resonance wavelength during target incubation that caused mostly by the larger bulk refractive index of the sample. After washing away the unbound target, the resonance wavelength shifts back to near the baseline wavelength. The net resonance wavelength shift is smaller than 0.04 nm, hence it is concluded that there was no detectable binding incident. Consequently, our detection method shows good specificity even by introducing high concentration of non-specific biomolecules.

Three protein biomarkers, HGF, MIP1, Fasligand were previously tested using ELISA at MD Anderson. To demonstrate that our sensors are able to detect different biomarkers that may be characteristic of pancreatic cancer, biosensing was also done for HGF and MIP1 which are known to be present at higher concentration in ELISA. Our tests detected the same proteins with 10 time dilution of the input sample. Figure 8 shows a 0.08 nm resonance wavelength shift with MIP1, and a 0.1 nm resonance wavelength shift of HGF. Fas ligand showed a 0.04 nm resonance wavelength shift which is our OSA detection limit. Thus the L13 PC microcavity without nanoholes was easily able to detect the HGF and MIP1 with a concentration of 9.813pg/ml and 15.437 pg/mL.

Since L13 PC microcavity without nanoholes can only detect the undiluted patient plasma sample with Fasligand biomarker, in order to achieve lower concentrations that ELISA cannot detect, the L13 PC microcavity with nanoholes is applied. In the case of L13 PC microcavity with nanoholes, the undiluted sample had a Fasligand concentration of 0.44 pM. Similar measurements were done as for L13 PC microcavity without nanoholes. As observed in Figure 9, compared to the L13 without nanoholes, a large resonance wavelength shift was observed for the 10 times sample dilution for the L13 PC microcavity with nanoholes ($R_{D}=0.4R$). Moreover, a measurable resonance wavelength shift of 0.08 nm, greater than the 0.04 nm detection limit of our system is observed for this sample with 50 times dilution. (8.8fM)(0.334 pg/mL). L13 and L13H are on the same chip and tested together, but only one (L13) device could be monitored in real-time. The other (L13H) only has discrete data points. Since Fasligand is present with the lowest concentration compared to HGF and MIP1 in the tested patient plasma, it is expected that with the L13 PC microcavity with nanoholes ($R_{D}=0.4R$) the detection can be easily achieved with two other biomarkers. It is worthwhile to contrast this detection sensitivity value with 1fM(67 fg/mL) of avidin binding to biotin in PBS in past research [28]. Table 3 compares the results from this research with measurements in ELISA. Binding changes the mass density on the waveguide surface and therefore the index refraction [33]. The resonant wavelength shift here are therefore observed experimentally. The heavier molecules lead to larger wavelength shift.

## 4. Conclusions

In conclusion, our experimental results show the potential of our lab-on-chip sensing device is capable of detecting plasma proteins in pancreatic cancer patient plasma with higher sensitivity than ELISA. Higher sensitivity is achieved enabling not only earlier detection but also renders to smaller sample volume requirement when compared with ELISA. Both are critical parameters in cancer biomarker research and in medical diagnostics. We detected biomarkers HGF and MIP1 present in the patient plasma having pancreatic cancer with concentration of 9.813 pg/mL and 15.437 pg/mL with a PC microcavity with L13 feature without nanoholes. Low concentration Fasligand antibody was detected with such a microcavity with nanoholes down to a concentration of 0.3346 pg/mL representing a 50 times lower concentration than that detected by ELISA on the same biomarker concentration. All devices were fabricated by standard CMOS foundry commercially available thereby justifying the capacity for high volume manufacturing.

## Figures and Tables

**Figure 1 micromachines-11-00282-f001:**
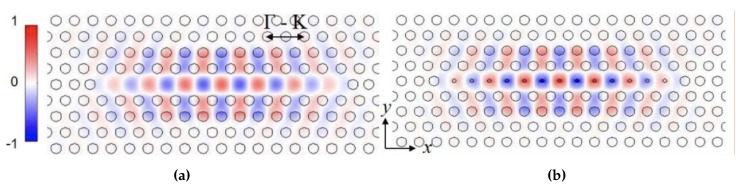
2D electric field intensity profile of the resonance mode in (**a**) L13 PC microcavity and (**b**) L13 PC microcavity with nanoholes.

**Figure 2 micromachines-11-00282-f002:**
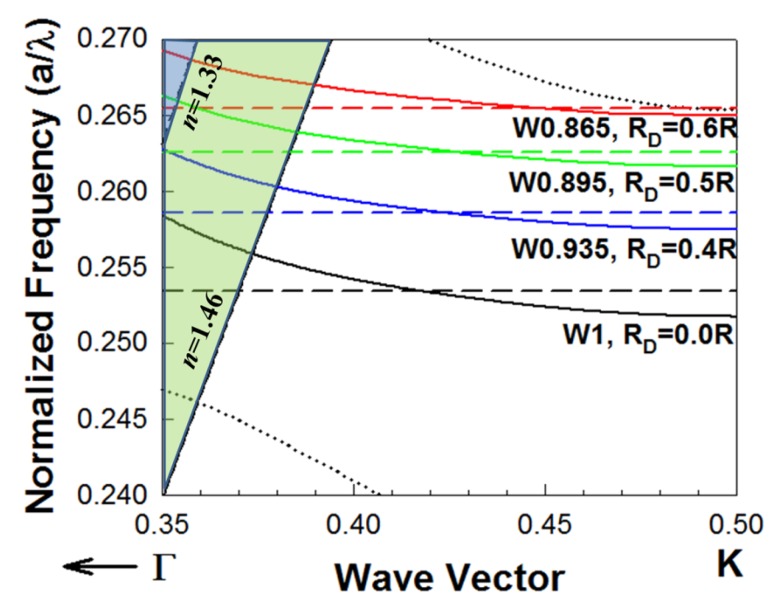
Dispersion diagram of two-dimensional PCWs in a triangular lattice PC etched into a silicon slab in silicon-on-insulator substrate in water ambient by 3D PWE. The dispersion of the guided mode for W1 (black), W0.935(blue), W0.895 (green) and W0.865 (red) PCWs are indicated in solid lines. For a W1 PCW, the coupled L13 PC microcavity does not have nanoholes ($R_{D}=0.0R$) and the resonance mode frequency is indicated by dashed black line. For L13 PC microcavities with nanoholes, the resonance frequencies for individual PC microcavities when $R_{D}$ = 0.4R, 0.5R and 0.6R are indicated by dashed blue, dashed green and dashed red lines respectively. The PCW slab guided mode has even parity. Dotted black lines denote the dielectric band (bottom) and odd parity (top) modes of the W1 PCW respectively, which are also raised in frequency (not shown in the figure) in the L13 nanohole cavity coupled PCWs. Light lines for water (n=1.33) and bottom silicon dioxide cladding (n=1.46) show the out-of-plane confinement of both the guided PCW mode and the confined microcavity mode below the light line.

**Figure 3 micromachines-11-00282-f003:**
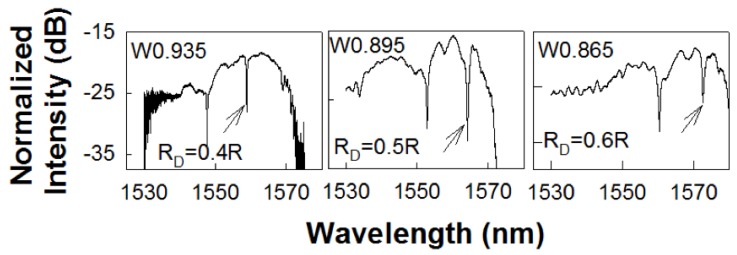
Transmission spectrum of a PC L13 nanohole cavity, with different nanohole size coupled to a PC waveguide with (**a**) $R_{D}=0.4R$, (**b**) $R_{D}=0.5R$ and (**c**) $R_{D}=0.6R$ respectively. The resonance mode of interest, closest to the transmission band edge of the coupling PCW, is indicated by black arrow.

**Figure 4 micromachines-11-00282-f004:**
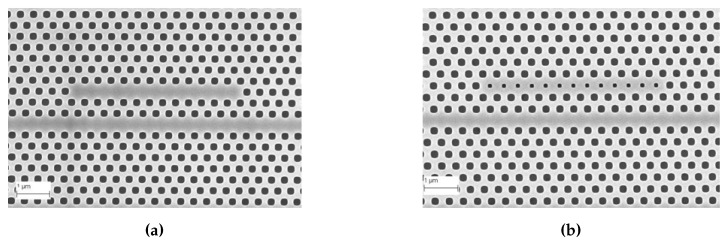
Images from Scanning Electron Microscope (**a**) L13 (i. e., 13 holes removed) PC microcavity and (**b**) L13 PC microcavity with nanoholes, fabricated by 193nm UV lithography in IMEC, Ghent, Belgium.

**Figure 5 micromachines-11-00282-f005:**
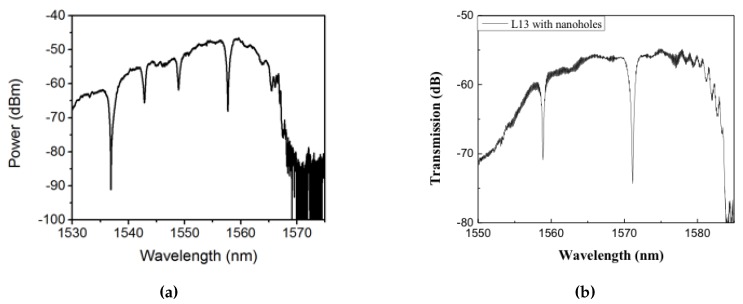
Optical spectrum at the output of the biochip showing sharp resonances in (**a**) L13 photonic crystal PC cavity and (**b**) L13 PC cavity with nanoholes corresponding to Figure 4.

**Figure 6 micromachines-11-00282-f006:**
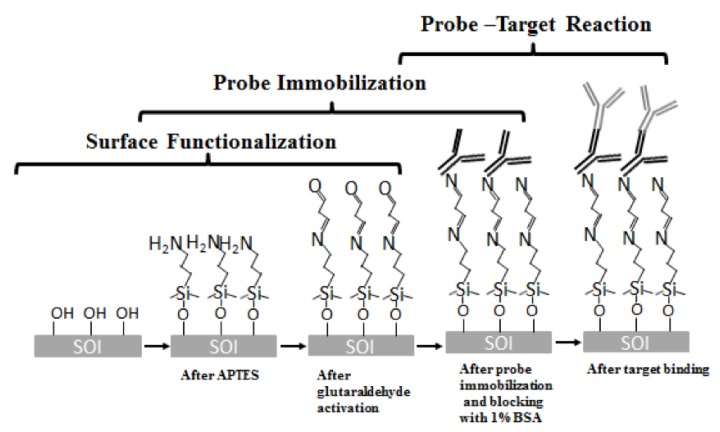
The binding chemical structure in biosensing reaction.

**Figure 7 micromachines-11-00282-f007:**
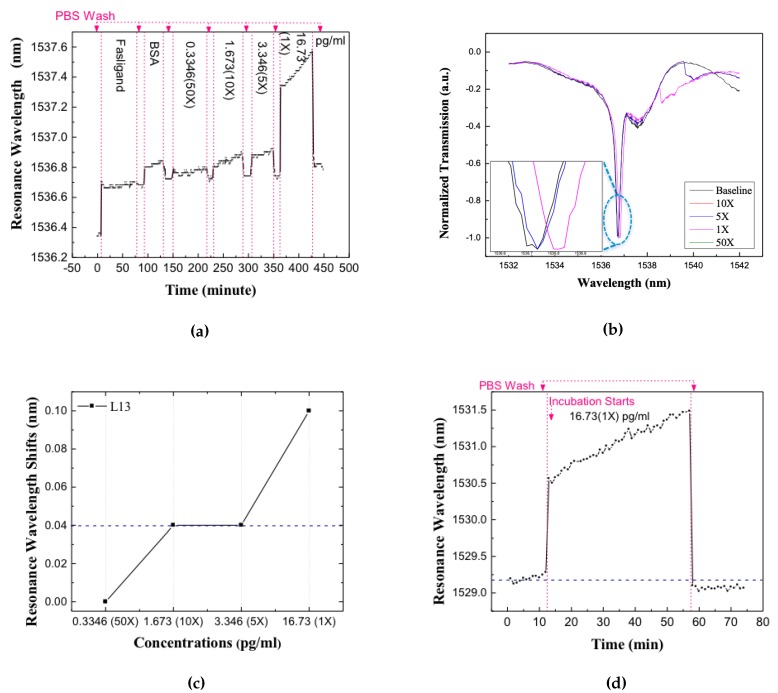
The detection result of L13 PC microcavity test for different dilutions concentrations of pancreatic patient plasma. (**a**) The real time resonance wavelength shift vs. time. Resonance wavelength increase during sample incubation and decrease after PBS wash step that wash away unbound molecules (**b**) Normalized spectra of the resonance after applying different concentrations of pancreatic patient plasma. (**c**) Resonance wavelength shifts for L13 in different dilutions, the blue dashed line is the optical spectrum analyzer detection limits. (**d**) Resonance wavelength shift vs. time in the specificity test, when introducing probe and target that are not the same conjugate pair, the blue dashed line indicates the origin baseline.

**Figure 8 micromachines-11-00282-f008:**
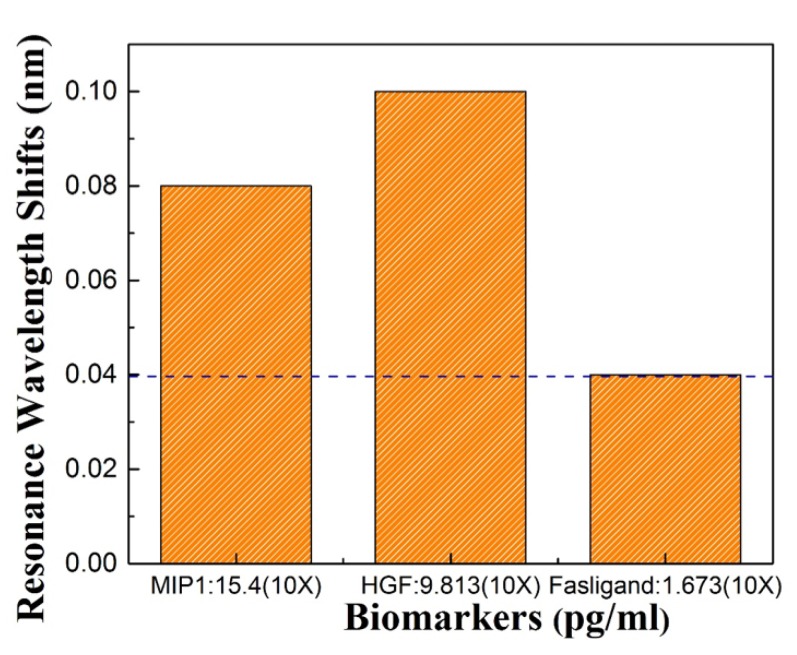
Resonance shift for the three biomarkers: Fasligand, MIP1, HGF to detect pancreatic patient plasma.

**Figure 9 micromachines-11-00282-f009:**
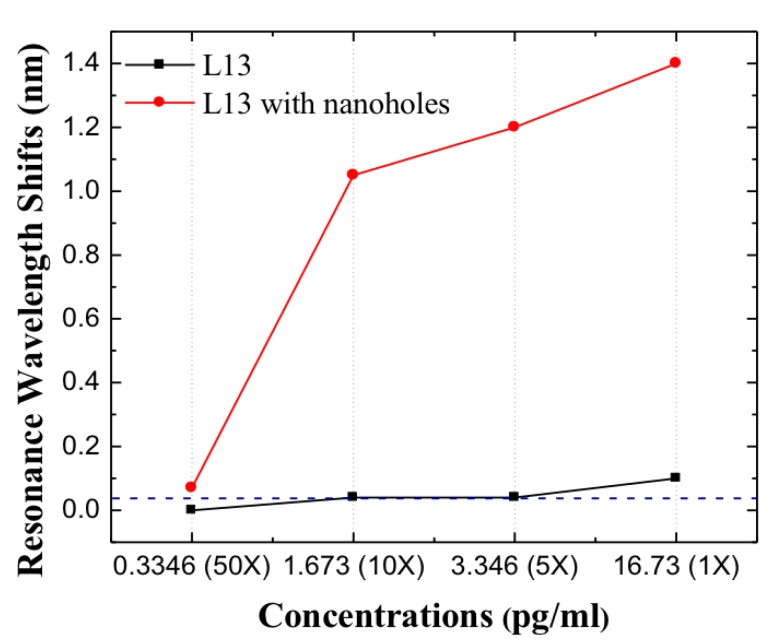
Resonance wavelength shifts for L13 and L13 with nanoholes in different dilutions, the blue dashed line is the optical spectrum analyzer detection limits.

**Table 1 micromachines-11-00282-t001:** The antibodies and chemical samples use in this paper.

Items	Company	Cat#
3-aminopropyl-triethoxy-silane (3-APTES)	Acros	919-30-2
Fasligand/CD178 Antibody (Fasligand)	Thermo Scientific	A5-32400
MacrophageInflammatory ProteinsAntibody (MIP1)	Thermo Scientific	710391
Hepatocyte Growth FactorAntibody (HGF)	Thermo Scientific	701283
Human Immunoglobulin G Antibody (Human IgG)	Abcam	Ab109489

**Table 2 micromachines-11-00282-t002:** The molecular weight of the antibody biomarkers and antigen.

	Antigen			Antibody		
	HGF	MIP1	Fasligand	HGF	MIP1	Fasligand
Molecular Weight (kD)	80	8	38	64	11	68

**Table 3 micromachines-11-00282-t003:** The detection result from PC microcavities in silicon compared to ELISA.

Biomarker	Our Detection Concentration (pg/mL)	ELISA Detection Concentration (pg/mL)
Macrophage Inflammatory Proteins Antibody (MIP1)	15.44 (L13)	3.9
Fasligand/CD178 Antibody (Fasligand)	0.334 (L13 with nanoholes)	6.44
Hepatocyte Growth Factor Antibody (HGF)	9.813 (L13)	47.72
Human Immunoglobulin G Antibody (Human IgG)	No reaction with our probe	Not applicable

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
