# Peer review of "Ultra Sensitivity Silicon-Based Photonic Crystal Microcavity Biosensors for Plasma Protein Detection in Patients with Pancreatic Cancer"

_micromachines, 2020, doi:10.3390/mi11030282_

Round 1

Reviewer 1 Report

The authors demonstrated photonic crystal microcavity (PCM) with nanoholes to enhance the detection of low concentration of Fasligand antibody. They first introduced the working principle of PCMs and the design factors in influencing the detection limits. Then, PCMs with different size of nanoholes and the dispersion diagram of coupled photonic crystal waveguides (PCWs) were studied. The transmission spectra of PCM coupled with waveguide devices were measured and applied to the detection of three different biomarkers in plasma from pancreatic cancer patients. By introducing the nanoholes to PCMs, low concentration of Fasligand antibody was detected around 50 times lower than that detected by ELISA.

First, several illustrations of PCM biosensors are not consistent and puzzled to the readers. For example, In Page 1 line 1: “Defect-engineered photonic crystal (PC) microcavities were fabricated by E-beam photolithography,” and page 2 line 64, “ …Devices were fabricated by using 193 nm UV photo-lithography,…” It is quite confused whether the PCMs were fabricated by E-beam lithography or UV photolithography? Or both methods have been used in the fabrication process? Second, Fig. 1(a) and (b) show the electric-field of PCMs without and with nanoholes, respectively. However, it is hard to tell the field distribution from this figure since the colorbar is missing. Also, the dispersion diagram shows the resonant frequency blueshifts when increasing the radius of nanoholes. However, the measured transmission dips show a contradictory trend where the resonant dips redshift for PCMs with larger size of nanoholes. Since the results are not well presented and the designs of biosensors in this paper are very similar to Ref. 27 (Applied physics letters 2014, 104, 191109.), I do not recommend publishing this article.

Author Response

It's by 193 nm UV photo-lithography, updated manuscript. Fig. 1(a) and (b) are updated in manuscript

Reviewer 2 Report

(1) By comparing the optical spectra in Figure 5, does the introduction of nanoholes causes disappearance of some small resonance peaks below 1550 nm, shift in the position of resonance peaks or raise a new peak at ~1570 nm? If so, how?

(2) What is the magnitude of shift in the position of resonance peaks and Q factor for L13 PC cavity with different RD/R ratio? Does RD=0.4R give the highest Q and high fb? It is not clearly written in the manuscript. It would be nice to see correlation graph between these parameters.

(3) Caption in Fig 3 mentioned a, b and c, but there is no labeling of a, b and c in the figure.

(3) Is it possible to explain the difference between Fig 3a and Fig 5b.

(4) Typo on Line 138. Fig 5b instead of Fig 5a.

(5) Some of the nomenclature and figure caption is unclear whether it is using L13 PC cavity with or without nanoholes. 

(5) Typo on Line 188. Treated instead of treateded.

Author Response

(1) The introduction of nanoholes reduces the optical length of the cavity and causes the distance between resonances to increase, so fewer resonances are observed.   This is similar to the longitudinal mode of a laser cavity, the smaller the cavity, the larger the longitudinal mode separation.

(2) I don’t think we have the correlation graph to show, but we do have some related analysis in a previous paper ref [27] which is cited in the paper.

(3)Explain the difference between Fig 3a and Fig 5b: These two spectra are from different batches. The difference could be from fabrication variations that causes the whole spectrum shift as we observed experimentally.  The beauty of the reported device is that it is irrelevant to the absolute of the wavelength itself.  The shift of the wavelength is the key process in this measurement.

Reviewer 3 Report

The authors shew the experimental quantitative detection of three different biomarkers of pancreatic cancer with label-free silicon photonic biosensor comprising of photonic crystal cavities. They compared their results with the conventional ELISA method. Introduction to the design and experimental process are delicate and the outcomes are impressive like the improved sensitivity of the cavity with nano-holes. However, I find some statements are not very clear.

What were the concentrations of the three biomarkers for ELISA in Table 3? Were they the same as that in the detections with the PC cavities? If the detection limit of ELISA can’t be determined, it is hard to say the PC cavity-based method outperforms the ELISA, since ELISA can detect a smaller concentration than the PC cavity-based method for MIPI. Is it needed to normalize the measured wavelength shift to the measurable limit of 0.04 nm? I mean, for the wavelength shift of 0.08 nm at 50X concentration, is the actual value exactly 0.08? For Figure 3, how did the author normalize the intensity? It seems the insertion loss is very high. Some minor typos and gramma errors:

P2,line31: suffer from

P2, line40: contribute to

P5, line128: Fig.4 and

P5, line137: Fig.5a and 5a

P9, line216: expected that with the L13

Author Response

What were the concentrations of the three biomarkers for ELISA in Table 3? Were they the same as that in the detections with the PC cavities? If the detection limit of ELISA can’t be determined, it is hard to say the PC cavity-based method outperforms the ELISA, since ELISA can detect a smaller concentration than the PC cavity-based method for MIPI.

Response to Reviewer:

concentrations of the three biomarkers for ELISA already shown on Table 3.

No, it’s the minimum concentration ELISA can detect, except MIP1, our method outperforms. 

Is it needed to normalize the measured wavelength shift to the measurable limit of 0.04 nm? I mean, for the wavelength shift of 0.08 nm at 50X concentration, is the actual value exactly 0.08? For Figure 3, how did the author normalize the intensity? It seems the insertion loss is very high.

Response to Reviewer:

We don’t count resonance shift smaller than 0.04 nm as real red shift.

This is explained in the paper (line 182 – 188)

Fig.7c  is at 0.04 nm  representing the noise floor of our bio detection system, mainly from the +/- 0.02 nm

  wavelength imprecision of our optical spectrum analyzer (OSA). In our L13 photonic crystal microcavity devices,

  the temperature dependence of wavelength shift is measured to be 0.08 nm / o C. During measurements, it was

  confirmed that the chip temperature is stabilized to within +/- 0.1_ C, the stage is thermally controlled with

  Newport 3040 Temperature Controller to avoid temperature induced resonance shift in the biosensors, thus the

  error from temperature is much lesser than our OSA detection error. Hence, only resonance wavelength shifts

  larger than 0.04 nm were treated as actual shifts caused by the specific binding between the probe and target.

Figure 3 is normalized from the transmission (dB) vs wavelength (nm) plot in a way so that the peak top and bottom extends between 0 and -1. The insertion loss comes from multiple sources including grating coupler, photonic crystal waveguide  (PCW), and water absorption. In this paper, we focus on the demonstration of resonance shift in response to bio-interactions, so insertion loss reduction is not optimized. However, further improvement can be made through optimized design and better fabrication control.

Round 2

Reviewer 1 Report

There are still two points not well explained in this revised version.

The dispersion diagram (Fig. 2) shows the resonant frequency increases when increasing the radius of nanoholes. However, the measured transmission dips (Fig. 3) show a contradictory trend where the indicated resonant dips redshift for PCMs with larger size of nanoholes.  The designs of biosensors in this paper are very similar to Ref. 27 (Applied physics letters 2014, 104, 191109.). Please clarify the novelty for this current work.

Author Response

The reviewer misunderstood the data.  No red or blue shift for the device itself when there is no biofluid carrying the markers interacted with the cavity.  The nanohole provides sensitivity enhancement clearly shown in figure 7 and figure 9 where nanohole device is much more sensitive than regular PCWs.  The innovation is to use this device detecting the pancreatic cancer biomarkers with a much higher sensitivity.